# An Analysis of SVD for Deep Rotation Estimation

**Jake Levinson**[1]     **Carlos Esteves**[2]    **Kefan Chen**[3]        **Noah Snavely**[3]
**Angjoo Kanazawa**[3]      **Afshin Rostamizadeh**[3]       **Ameesh Makadia**[3]
[1]Simon Fraser University      [2]University of Pennsylvania      [3]Google Research

## Abstract

Symmetric orthogonalization via SVD, and closely related procedures, are well-known techniques for projecting matrices onto $O(n)$ or $SO(n)$. These tools have long been used for applications in computer vision, for example optimal 3D alignment problems solved by orthogonal Procrustes, rotation averaging, or Essential matrix decomposition. Despite its utility in different settings, SVD orthogonalization as a procedure for producing rotation matrices is typically overlooked in deep learning models, where the preferences tend toward classic representations like unit quaternions, Euler angles, and axis-angle, or more recently-introduced methods. Despite the importance of 3D rotations in computer vision and robotics, a single universally effective representation is still missing. Here, we explore the viability of SVD orthogonalization for 3D rotations in neural networks. We present a theoretical analysis of SVD as used for projection onto the rotation group. Our extensive quantitative analysis shows simply replacing existing representations with the SVD orthogonalization procedure obtains state of the art performance in many deep learning applications covering both supervised and unsupervised training.

## 1   Introduction

There are many ways to represent a 3D rotation matrix. But what is the ideal representation to predict 3D rotations in a deep learning framework? The goal of this paper is to explore this seemingly low-level but practically impactful question, as currently the answer appears to be ambiguous.

In this paper we present a systematic study on estimating rotations in neural networks. We identify that the classic technique of SVD orthogonalization, widely used in other contexts but rarely in the estimation of 3D rotations in deep networks, is ideally suited for this task with strong empirical and theoretical support.

3D rotations are important quantities appearing in countless applications across different fields of study, and are now especially ubiquitous in learning problems in 3D computer vision and robotics. The task of predicting 3D rotations is common to estimating object pose [53, 27, 32, 44, 49, 24, 45], relative camera pose [30, 36, 7], ego-motion and depth from video [55, 29], and human pose [56, 21].

A design choice common to all of these models is selecting a representation for 3D rotations. The most frequent choices are classic representations including unit quaternion, Euler angles, and axis-angle. Despite being such a well-studied problem, there is no universally effective representation or regression architecture due to performance variations across different applications.

A natural alternative to these classic representations is symmetric orthogonalization, a long-known technique which projects matrices onto the orthogonal group $O(3)$ [26, 40]. Simple variations can restrict the projections onto the special orthogonal (rotation) group $SO(3)$ [15, 20, 50]. This procedure, when executed by Singular Value Decomposition (SVD [11]), has found many applications

---

contact: `jake_levinson@sfu.ca`, `makadia@google.com`

in computer vision, for example at the core of the Procrustes problem [2, 40] for point set alignment, as well as single rotation averaging [13]. A nearly identical procedure is used for factorizing Essential matrices [14].

Despite its adoption in these related contexts, orthogonalization via SVD has not taken hold as a procedure for generating 3D rotations in deep learning: it is rarely used when implementing a model (e.g. overlooked in [24, 7, 36, 29]), nor is it considered a benchmark when evaluating new representations [25, 57, 35].

In light of this, this paper explores the viability of SVD orthogonalization for estimating rotations in deep neural networks. Note, we do not claim to be the first to introduce this tool to deep learning, rather our focus is on providing a comprehensive study of the technique specifically for estimating rotations. Our contributions include

- A theoretically motivated analysis of rotation estimation via SVD orthogonalization in the context of neural networks, and in comparison to the recently proposed Gram-Schmidt procedure [57]. One main result is that SVD improves over Gram-Schmidt by a factor of two for reconstruction, thus supporting SVD as the preferred orthogonalization procedure.

- An extensive quantitative evaluation of SVD orthogonalization spanning four diverse application environments: point cloud alignment, object pose from images, inverse kinematics, and depth prediction from images, across supervised and unsupervised settings, and benchmarked against classic and recently introduced rotation representations.

Our results show that rotation estimation via SVD orthogonalization achieves state of the art performance in almost all application settings, and is the best performing method among those that can be applied in both supervised and unsupervised settings. This is an important result given the prevalence of deep learning frameworks that utilize rotations, as well as for benchmarking future research into new representations.

## 2   Related Work

Optimization on $SO(3)$, and more generally on Riemannian manifolds, is a well-studied problem. Peculiarities arise since $SO(3)$ is not topologically homeomorphic to any subset of 4D Euclidean space, so any parameterization in four or fewer dimensions will be discontinuous (this applies to all classic representations—Euler angles, axis-angle, and unit quaternions). Discontinuities and singularities are a particular nuisance for classic gradient-based optimization on the manifold [43, 47].

Early deep learning models treated Euler angle estimation as a classification task [49, 44], by discretizing the angles into bins and using softmax to predict the angles. This idea was extended to hybrid approaches that combine classification and regression. In [23], discrete distributions over angles are mapped to continuous angles via expectation and [28] combines classification over quantized rotations with the regression of a continuous offset. In [25] it is shown that typical activations used in classification models (e.g. softmax) lead to more stable training compared to the unconstrained setting of regression. The authors introduce a "spherical exponential" mapping to bridge the gap and improve training stability for regression to $n$-spheres. All of the above methods require supervision on the classification objective, which makes them unsuitable for unsupervised settings.

Probabilistic representations have been introduced for modeling orientation with uncertainty [38, 10], with the von Mises and Bingham distributions respectively. While these are best suited for multi-modal and ambiguous data, such approaches do not reach state of the art in tasks where a single precise rotation must be predicted.

The closest approach to SVD orthogonalization is the recent work of [57] which makes a strong connection between the discontinuities of a representation and their effect in neural networks. In search of continuous representations, they propose the idea of continuous overparameterizations of $SO(3)$, followed by continuous projections onto $SO(3)$. Their $6D$ representation is mapped onto $SO(3)$ via a partial Gram-Schmidt procedure. This is similar in spirit to SVD orthogonalization which will map a continuous 9D representation onto $SO(3)$ with SVD. We leave the deeper comparison of these two methods to the following sections, where we show that SVD provides a more robust projection onto $SO(3)$.

A common application for optimization on $SO(3)$ is state estimation in robotics, and the relevant derivatives (e.g. Jacobian of the logarithmic map) have been analyzed [3, 42]. In our setting we must consider SVD derivatives, which have been presented in [9, 33]. There exist multiple works that build neural nets with structured layers depending on SVD or Eigendecomposition [18, 37, 17, 5], showing SVD is amenable for learning via backpropagation. The closest to our setting is [46] which applies the orthogonal Procrustes problem to 3D point set alignment within a neural network, and [19] which proposes singular value clipping to regularize networks' weight matrices. We discuss the stability of SVD orthogonalization in neural networks in the following section.

## 2.1 Contemporaneous works

In [35], the quaternion form of the Wahba alignment problem is considered [50, 54]. The unit quaternion that best aligns two point sets can be computed via the eigendecomposition of a symmetric data matrix, and the proposed network model regresses directly the elements of this 4x4 symmetric matrix.

In [31] the network regresses the parameters of a matrix Fisher distribution [22]. For training, the distribution's non-trivial normalizing constant and its gradient are approximated. For inference, the mode of the distribution can be computed using the same SVD orthogonalization we analyze in this work.

## 3 Analysis

In this section we present a theoretically motivated analysis of SVD orthogonalization for rotation estimation. We start here defining the procedures and introducing well-known results regarding their least-squares optimality before presenting the analysis.

Given a square matrix $M$ with SVD $U\Sigma V^T$, we consider the orthogonalization and *special* orthogonalization

$$\texttt{SVDO}(M) := UV^T, \tag{1}$$

$$\texttt{SVDO}^+(M) := U\Sigma'V^T, \text{ where } \Sigma' = \text{diag}(1,\ldots,1,\det(UV^T)). \tag{2}$$

$\texttt{SVDO}$ is orientation-preserving, while $\texttt{SVDO}^+$ maps to $SO(n)$. Orthogonalization of a matrix via SVD is also known as *symmetric orthogonalization* [26]. It is well known that symmetric orthogonalization is optimal in the least-squares sense [2, 15, 40]:

$$\texttt{SVDO}(M) = \underset{R\in O(n)}{\arg\min}\|R - M\|_F^2, \qquad \texttt{SVDO}^+(M) = \underset{R\in SO(n)}{\arg\min}\|R - M\|_F^2. \tag{3}$$

This property has made symmetric orthogonalizations useful in a variety of applications [50, 40, 46]. To be specific, $\texttt{SVDO}^+(M)$ is the procedure we will evaluate experimentally in Section 4 for 3D rotation estimation in neural networks.

## 3.1 $\texttt{SVDO}(M)$ and $\texttt{SVDO}^+(M)$ are maximum likelihood estimates

From Eq. 3 it follows that SVD orthogonalization maximizes the likelihood in the presence of Gaussian noise. Let $M = R_\mu + \sigma N$ represent an observation of $R_\mu \in SO(n)$, corrupted by noise $N$ with entries $n_{ij} \sim \mathcal{N}(0, 1)$. With the matrix normal pdf [12], the likelihood function is

$$L(R_\mu; M, \sigma) = ((2\pi)^{\frac{n^2}{2}} \sigma^{n^2})^{-1} \exp(-\tfrac{1}{2\sigma^2}((M - R_\mu)^T(M - R_\mu))). \tag{4}$$

$L(R_\mu; M, \sigma)$, subject to $R_\mu \in SO(n)$, is maximized when $(M - R_\mu)^T(M - R_\mu)$ is minimized:

$$\underset{R_\mu\in SO(n)}{\arg\max} L(R_\mu; M, \sigma) = \underset{R_\mu\in SO(n)}{\arg\min} (M - R_\mu)^T(M - R_\mu) = \underset{R_\mu\in SO(n)}{\arg\min} \|M - R_\mu\|_F^2 \tag{5}$$

The minimum is given by $\texttt{SVDO}^+(M)$ (Eq. 3), and similarly by $\texttt{SVDO}(M)$ when $R_\mu \in O(n)$.

## 3.2 Gradients

In this section we analyze the behavior of the gradients of a network with an $\texttt{SVDO}^+$ layer, and show that they are generally well behaved. Specifically, we can consider $\frac{\partial L}{\partial M}$ for some loss function

$L(M, R) = \|\text{SVDO}^+(M) - R\|_F^2$. We will first analyze $\frac{\partial L}{\partial M}$ for $\text{SVDO}(M)$. Letting $\circ$ denote the Hadamard product, from [18, 48] we have

$$\frac{\partial L}{\partial M} = U[(F^T \circ (U^T \frac{\partial L}{\partial U} - \frac{\partial L}{\partial U}^T U))\Sigma + \Sigma(F^T \circ (V^T \frac{\partial L}{\partial V} - \frac{\partial L}{\partial V}^T V))]V^T, \quad (6)$$

$$F_{i,j} = \begin{cases} \frac{1}{s_i^2 - s_j^2}, & \text{if } i \neq j \\ 0, & \text{if } i = j \end{cases}, \quad s_i = \Sigma_{ii} \quad (7)$$

Letting $X = U^T \frac{\partial L}{\partial U} - \frac{\partial L}{\partial U}^T U$, we can simplify $\frac{\partial L}{\partial M} = UZV^T$ where the elements of $Z$ are

$$Z_{ij} = \begin{cases} \frac{X_{ij}}{s_i + s_j}, & \text{if } i \neq j \\ 0, & \text{if } i = j. \end{cases} \quad (8)$$

For $\text{SVDO}(M)$, Eq. 8 tells us $\frac{\partial L}{\partial M}$ is undefined whenever two singular values are both zero and large when their sum is very near zero. In this case $\frac{\partial L}{\partial M}$ is undefined if the smallest singular value occurs with multiplicity greater than 1. It is large if the two smallest singular values are close to each other, or if they are close to 0. See Section C in the supplement for the detailed derivations.

### 3.3 Error analysis

In this section we approximate the expected error in $\text{SVDO}(M)$ and Gram-Schmidt orthogonalization (denoted as $\text{GS}(M)$) in the presence of Gaussian noise, and observe that the error is twice as large for $\text{GS}$ as for $\text{SVDO}$. The noise model represents errors in unconstrained network outputs rather than errors on $SO(3)$, thus we use a Gaussian model rather than one appropriate for $SO(3)$, such as Bingham [4] or Langevin [39]. If $M$ is a matrix with QR decomposition $M = QR$, define:

$$\text{GS}(M) := Q, \qquad \text{GS}^+(M) := Q\Sigma'', \text{ where } \Sigma'' = \text{diag}(1, \ldots, 1, \det(Q)). \quad (9)$$

We consider $M = R_0 + \sigma N$, a noisy observation of a rotation matrix $R_0 \in SO(n)$, where $N$ has i.i.d. Gaussian entries $n_{ij} \sim \mathcal{N}(0, 1)$ and $\sigma$ is small. The analysis is independent of $R_0$, so for simplicity we set $R_0 = I$. First we calculate the SVD and QR decompositions of $M$ to first order for $N$ an arbitrary (non-random) matrix.

**Proposition 1** *The SVD and QR decompositions of $M = I + \sigma N$ are as follows:*

1. *(SVD) Let $N = S + A$ be the decomposition of $N$ into symmetric and antisymmetric parts. Then, to first order, an SVD of $M$ is given by*

$$M = U_0(I + \sigma U_1) \cdot (I + \sigma \Sigma_1) \cdot (I + \sigma V_1)^T U_0^T,$$

*where $U_0 \Sigma_1 U_0^T$ is an SVD of $S$, and $U_1, V_1$ are (non-uniquely determined) antisymmetric matrices satisfying $U_0^T A U_0 = U_1 + V_1^T$.*

2. *(QR) Let $N = U + D + L$ be the strict upper-triangular, diagonal, and strict lower-triangular parts of $N$. To first order, $M$ has QR decomposition*

$$M = (I + \sigma Q_1) \cdot (I + \sigma R_1),$$

*where $Q_1 = L - L^T$ and $R_1 = D + U + L^T$.*

*Consequently, $\text{SVDO}(M) = I + \sigma A + O(\sigma^2)$ and $\text{GS}(M) = I + \sigma(L - L^T) + O(\sigma^2)$.*

**Corollary 1** *If $N$ is 3x3 with i.i.d. Gaussian entries $n_{ij} \sim \mathcal{N}(0, 1)$, then with error of order $O(\sigma^3)$,*

$$\mathbb{E}[\|\text{SVDO}(M) - I\|_F^2] = 3\sigma^2, \qquad \mathbb{E}[\|\text{GS}(M) - I\|_F^2] = 6\sigma^2 \quad (10)$$

$$\mathbb{E}[\|\text{SVDO}(M) - M\|_F^2] = 6\sigma^2, \qquad \mathbb{E}[\|\text{GS}(M) - M\|_F^2] = 9\sigma^2 \quad (11)$$

See Section A in the supplement for the proofs. Notably, Gram-Schmidt produces *twice* the error in expectation (and indeed deviates 1.5 times further from the observation $M$ itself). The same holds for

SVDO$^+$ and GS$^+$: the probability that $\det(M) < 0$ decays exponentially (i.e. faster than polynomially) as $\sigma \to 0$, so any finite-order error analysis is identical for SVDO$^+$ and GS$^+$. The difference in performance between SVDO and GS can be traced to the fact that Gram-Schmidt is essentially "greedy" with respect to the starting matrix, whereas the SVD approach is coordinate-independent.

Although i.i.d. Gaussian noise is not necessarily reflective of a neural network's predictions, it does provide insight into the relationship between SVDO$^+$ and GS$^+$. See Section A in the supplement for further remarks.

### 3.4 Continuity for special orthogonalization

The calculation above shows SVDO$(M)$ and SVDO$^+(M)$ are continuous and differentiable, at least at $M = I$. In fact SVDO$(M)$ is smooth, as is SVDO$^+$ except for a discontinuity[1] if (and only if) $\det(M) = 0$ or $\det(M) < 0$ and its smallest singular value has multiplicity greater than 1. In fact the optimization problem (3) is degenerate in this case. For example, the 2×2 matrix $M = \mathrm{diag}(1, -1)$ is equidistant from every rotation matrix; perturbations of $M$ may special-orthogonalize to any $R \in SO(2)$. GS$^+$ is continuous on a slightly larger domain – $\det(M) \neq 0$ – because it makes a uniform choice, negating the $n$-th column of $M$ if necessary, at the cost of significantly greater error in expectation. This reflects the fact that SVD orthogonalization is coordinate-independent and GS, GS$^+$ are not:

$$\text{SVDO}(R_1 M R_2) = R_1 \text{SVDO}(M) R_2, \text{ for all } R_1, R_2 \in SO(n), M \in GL(n), \tag{12}$$

and similarly for SVDO$^+$. GS and GS$^+$ are rotation-equivariant on only one side: GS$(R_1 M) = R_1$GS$(M)$, but GS$(MR_2)$ is not a function of $R_2$ and GS$(M)$; likewise for GS$^+$. See Section B in the supplement for a proof of smoothness and further discussion.

### 3.5 Summary

The results above illustrate a number of desirable properties of SVD orthogonalization. It is well known that SVDO$^+$ is optimal in the least squares sense, as well as in the presence of Gaussian noise (MLE). We show that viewed through the lens of matrix reconstruction, the approximation errors are half that of the Gram-Schmidt procedure. Finally, we present the conditions that lead to large gradient norms (conditions that are rare for small matrices). In the following, we support this theoretical analysis with extensive quantitative evaluations.

## 4 Experiments

Recall, the SVD orthogonalization procedure SVDO$^+(M)$ takes a 9D network output (interpreted as a 3x3 matrix), and projects it onto $SO(3)$ via Eq. 2. The procedure can easily be used in popular deep learning libraries (e.g. PyTorch [34] and TensorFlow [1] both provide differentiable SVD ops). We did not notice an increase in training time with SVDO$^+(M)$ for most experiments as the pose layer is not the bottleneck.

**Methods.** Now we provide a short description of the methods under comparison (see Section D.1 in the supplement for further details). **SVD-Train** is SVDO$^+(M)$ (Eq. 2). **SVD-Inference** is SVDO$^+(M)$, except the training loss is applied directly to $M$. Since SVDO$^+$ is applied only at inference, it is a continuous representation for training. **6D** and **5D** are introduced in [57] for projecting 6D and 5D representations onto $SO(3)$. 6D is the partial Gram-Schmidt method which computes GS$^+(M)$ (Eq. 9), and 5D utilizes a stereographic projection. Our implementations follow the code provided by [57]. **QCQP** [35] is a contemporaneous method which predicts quaternions through the eigen-decomposition of a symmetric matrix: a network regresses the 10 parameters of a 4x4 symmetric matrix, and the predicted unit quaternion is given by the eigenvector of the smallest eigenvalue. The training loss is determined after mapping the quaternion to a rotation matrix. **Spherical Regression** [25] ($S^2$-**Reg**) regresses to $n$-spheres. The method combines regression to the absolute values of a unit quaternion with classification of the signs. We select the hyperparameter that balances the classification and regression losses by a simple line search in the neighborhood of the default provided [25]. **3D-RCNN** [23] combines likelihood estimation and regression (via expectation) for

Table 1: **3D point cloud alignment.** Left: a comparison of methods by *mean*, *median*, and *standard deviation* of (geodesic) errors after 2.6M training steps. Middle: mean test error at different points along the training progression. Right: test error percentiles after training completes. The legend on the right applies to both plots.

| | Mean (°) | Med | Std |
|---|---|---|---|
| 3D-RCNN | 5.51 | 1.91 | 17.05 |
| $M_G$ | 9.12 | 7.65 | 10.46 |
| Euler | 11.04 | 6.23 | 15.56 |
| Axis-Angle | 6.65 | 4.06 | 11.47 |
| Quaternion | 5.48 | 3.19 | 11.03 |
| $S^2$-Reg | 4.80 | 3.00 | 9.27 |
| 5D | 3.77 | 2.19 | 8.70 |
| 6D | 2.24 | 1.22 | 7.83 |
| QCQP | 1.90 | 1.07 | 6.77 |
| SVD-Inf | 2.64 | 1.60 | 8.16 |
| SVD-Train | **1.63** | **0.89** | 6.70 |

predicting Euler angles. This representation also requires both classification and regression losses for training. **Geodesic-Bin-and-Delta** ($M_G$ [28]) presents a hybrid model which combines classification over a quantized pose space with regression of offsets from the quantized poses. **Quaternion**, **Euler angles**, and **axis-angle** are the classic parameterizations. In each case they are converted to matrix form before the loss is applied to stay consistent with the experimental settings in [57].

For SVD, 6D, 5D, QCQP, and the classic representations, the loss is $L(R, R_t) = \frac{1}{2} \|R - R_t\|_F^2$. When $R, R_t \in SO(3)$ this is related to geodesic angle error $\theta$ as $L(R, R_t) = 2 - 2\cos(\theta)$. All other methods require an additional classification loss. See the supplement for additional experiments and details.

## 4.1 3D point cloud alignment

The first experiment is the point cloud alignment benchmark from [57]. Given two shape point clouds the network is asked to predict the 3D rotation that best aligns them. The rotations in the dataset are sampled uniformly from $SO(3)$ (no rotation bias in the data). Table 1 (left) shows geodesic error statistics (mean, median, std) on the test set. We omit reporting the maximum error as it is near $180°$ for all methods and cannot be attributed exclusively to the choice of representation, since errors are also due to limitations of model generalization to unseen (and sometimes almost symmetric) data. SVD-Train outperforms all the baselines. Interestingly, the hybrid approaches 3D-RCNN and $M_G$ underperform the top regression baselines, a point we will return to later. Table 1 (middle) shows the mean errors on the test set as training progresses. The best performing methods (SVD variations, QCQP, and 6D) also show fast convergence. The errors at different percentiles are shown in Table 1 (right).

Our choices for classic representation baselines are those which are most commonly used in deep learning architectures. The Cayley transform, which appears less frequently in these settings, had a mean and median error of $9.16°$ and $5.02°$ for supervised point cloud alignment, which is in the range of the other classic baselines.

The model architecture follows the architecture described in [57]. Point clouds are embedded with simplified PointNet (4-layer MLP) ending with a global max-pooling. Three dense layers make up the regression network. The output dimensionality of the final layer depends on the chosen representation. For the hybrid classification+regression models, the final layers follow the details provided in the relevant references.

In the supplement we show results for different experimental settings (training with different initial learning rates, with learning rate decay, geodesic loss, and rotations restricted to a subspace of $SO(3)$). The relative performances of the different methods, and specifically the effectiveness of SVD-Train, remains.

## 4.2 3D Pose estimation from 2D images

The second experiment follows the benchmark set forth in [25]. Images are rendered from Model-Net10 [51] objects from arbitrary viewpoints. Given a 2D image, the network must predict the object orientation. We used MobileNet [16] to generate image features, followed by the same fully connected

Table 2: **Pose estimation from ModelNet chair images.** We report the same metrics as in Table 1, see the caption there for a description. All models are trained for 550K steps in this case.

| | Mean (°) | Med | Std |
|---|---|---|---|
| 3D-RCNN | 35.50 | 13.21 | 46.55 |
| $M_G$ | 31.60 | 16.70 | 41.86 |
| Euler | 41.35 | 27.44 | 37.73 |
| Axis-Angle | 32.30 | 19.74 | 34.70 |
| Quaternion | 26.92 | 14.39 | 32.92 |
| $S^2$-Reg | 27.36 | 15.41 | 33.17 |
| 5D | 25.18 | 13.40 | 32.10 |
| 6D | 22.60 | 11.51 | 31.24 |
| QCQP | **20.57** | **10.76** | 29.38 |
| SVD-Inf | 21.38 | 11.41 | 29.35 |
| SVD-Train | 21.25 | 11.14 | 30.28 |

Table 3: **Pose estimation from ModelNet sofa images.** We report the same metrics as in Table 1, see the caption there for a description. All models are trained for 550K steps in this case.

| | Mean (°) | Med | Std |
|---|---|---|---|
| 3D-RCNN | 34.80 | 7.32 | 55.73 |
| $M_G$ | 31.41 | 13.93 | 48.48 |
| Euler | 49.31 | 32.03 | 43.47 |
| Axis-Angle | 31.82 | 17.31 | 37.19 |
| Quaternion | 29.60 | 14.56 | 37.00 |
| $S^2$-Reg | 25.99 | 12.11 | 37.67 |
| 5D | 26.23 | 11.52 | 38.91 |
| 6D | 20.25 | 7.84 | 36.85 |
| QCQP | 18.51 | 7.52 | 34.39 |
| SVD-Inf | 20.30 | 8.85 | 33.88 |
| SVD-Train | **18.01** | **7.31** | 33.96 |

regression layers as above. We found negligible difference in performance between MobileNet and VGG16 [41] (not surprising given the comparison in [16]), so we used MobileNet due to the reduced training time. Rather than averaging over all 10 ModelNet categories as in [25], we focus on *chair* and *sofa* which are the two categories which exhibit the least rotational symmetries in the dataset. Results are shown in Tables 2 and 3. Interestingly, SVD-Inference also performs similarly to SVD-Train on final metrics with faster convergence, indicating short pretraining with SVD-Inference could improve convergence rates.

3D-RCNN and $M_G$ are again underperforming the best methods. These hybrid methods have shown state of the art performance on predicting 3D pose from images [23, 28], but in those benchmarks the 3D rotations exhibit strong viewpoint bias (camera viewpoints are not evenly distributed over $SO(3)$). In our experiments so far, we have only considered random rotations uniformly sampled from $SO(3)$, This can explain the gap in performance and highlights a limitation of these hybrid methods.

## 4.3 Pascal 3D+

Pascal3D+ [52] is a standard benchmark for object pose estimation from single images. The dataset is composed of real images covering 12 categories. Following common experimental settings for this benchmark, for training we discard occluded or truncated objects [28, 44] and augment with rendered images from [44]. The model architecture is the same as in Section 4.2. Table 4 shows results on two categories and the mean over all categories (see Section D.5 in the supplement for results on each of the 12 categories). The individual metrics we report are the median error as well as accuracies at $10°$, $15°$, and $20°$.

The best performing method is clearly $S^2$-Regression. As expected, the hybrid method 3D-RCNN performs well on this task, but SVD-Inference and SVD-Train are on par. The SVD variations are also the best performing of the regression methods (those that only train with a rotation loss). Interestingly, SVD-Inference slightly outperforms SVD-Train, which suggests in this scenario where viewpoints have a non-uniform prior, training a network to regress directly to the desired target rotation matrix can work well when combined with SVD orthogonalization at inference.

Table 4: **Pascal 3D+.** Accuracy at $10°$, $15°$, and $20°$ (higher is better), and median error are reported. On the left are results for *sofa* and *bicycle*. The third block is the results averaged over all 12 categories, and these numbers are used to determine the ranks shown on the right (lower is better).

| | Sofa | | | | Bicycle | | | | Mean (12 categories) | | | | Rank (12 categories) | | | |
|---|---|---|---|---|---|---|---|---|---|---|---|---|---|---|---|---|
| | Accuracy@ | | | Med° | Accuracy@ | | | Med° | Accuracy@ | | | Med° | Accuracy@ | | | Med° |
| | $10°$ | $15°$ | $20°$ | Err | $10°$ | $15°$ | $20°$ | Err | $10°$ | $15°$ | $20°$ | Err | $10°$ | $15°$ | $20°$ | Err |
| 3D-RCNN | 37.1 | 54.3 | 80.0 | 14.2 | 17.8 | 38.6 | 72.3 | 16.9 | 43.2 | 57.6 | 78.1 | 12.9 | 2 | 3 | 6 | 2 |
| $M_G$ | 31.4 | 51.4 | 74.3 | 14.4 | 11.9 | 31.7 | 66.3 | 20.9 | 32.9 | 52.4 | 77.0 | 14.7 | 6 | 5 | 8 | 5 |
| Euler | 22.9 | 45.7 | 77.1 | 16.3 | 9.9 | 20.8 | 68.3 | 23.4 | 24.5 | 42.0 | 71.9 | 19.2 | 10 | 11 | 11 | 11 |
| Axis-Angle | 11.4 | 40.0 | 80.0 | 16.3 | 13.9 | 31.7 | 70.3 | 21.3 | 23.0 | 44.3 | 76.9 | 17.7 | 11 | 9 | 9 | 10 |
| Quaternion | 34.3 | 62.9 | 77.1 | 11.7 | 15.8 | 30.7 | 67.3 | 22.4 | 34.2 | 51.6 | 78.0 | 15.1 | 5 | 7 | 7 | 6 |
| $S^2$-Reg | 37.1 | **65.7** | 85.7 | 11.2 | **21.8** | **45.5** | 75.2 | **16.1** | **45.8** | **64.4** | **83.8** | **11.3** | **1** | **1** | **1** | **1** |
| 5D | 17.1 | 54.3 | 77.1 | 14.2 | 10.9 | 26.7 | 68.3 | 21.1 | 25.2 | 43.9 | 75.6 | 17.0 | 9 | 10 | 10 | 9 |
| 6D | 34.3 | 54.3 | **88.6** | 13.3 | 14.9 | 27.7 | 71.3 | 22.0 | 32.6 | 51.1 | 81.1 | 15.2 | 7 | 8 | 3 | 7 |
| QCQP | 42.9 | 54.3 | 82.9 | 13.7 | 5.0 | 18.8 | 66.3 | 21.8 | 31.8 | 51.6 | 80.0 | 15.3 | 8 | 6 | 5 | 8 |
| SVD-Inf | **45.7** | 60.0 | **88.6** | **11.0** | 10.9 | 33.7 | **84.2** | 19.0 | 39.9 | 58.7 | 83.7 | 13.0 | 3 | 2 | 2 | 3 |
| SVD-Train | 40.0 | 57.1 | 85.7 | 12.7 | 9.9 | 26.7 | 80.2 | 20.9 | 35.1 | 52.7 | 80.5 | 14.6 | 4 | 4 | 4 | 4 |

Table 5: **Self-supervised 3D point cloud alignment.** The error metrics presented follow the same format as the earlier supervised point cloud alignment experiment, see Table 1. Although here the model is trained without rotation supervision, we show test errors in the predicted rotations. The legend on the right applies to both plots.

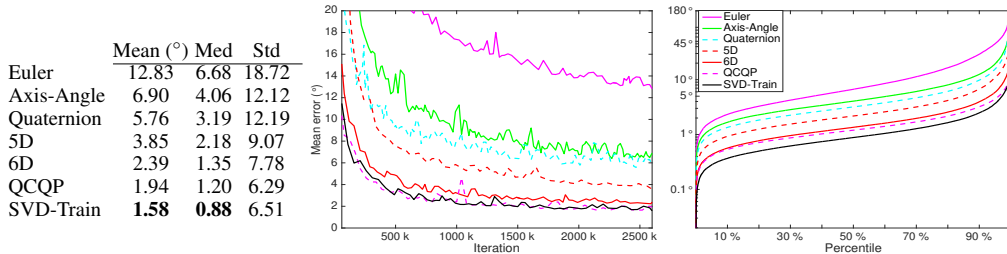

| | Mean (°) | Med | Std |
|---|---|---|---|
| Euler | 12.83 | 6.68 | 18.72 |
| Axis-Angle | 6.90 | 4.06 | 12.12 |
| Quaternion | 5.76 | 3.19 | 12.19 |
| 5D | 3.85 | 2.18 | 9.07 |
| 6D | 2.39 | 1.35 | 7.78 |
| QCQP | 1.94 | 1.20 | 6.29 |
| SVD-Train | **1.58** | **0.88** | 6.51 |

## 4.4 Unsupervised rotations

So far we have considered supervised rotation estimation. Given the shift towards self- or unsupervised 3D learning [55, 29, 21, 57], it is important to understand how different representations fare without direct rotation supervision. We omit 3D-RCNN, $M_G$, and $S^2$-Reg from the experiments below as they require explicit supervision of classification terms, as well as SVD-Inference as it does not produce outputs on $SO(3)$ while training.

### 4.4.1 Self-supervised 3D point cloud alignment

To begin, we devise a simple variation of the point cloud alignment experiment from Section 4.1. Given two point clouds, the network still predicts the relative rotation. However, now the only loss is an L2-loss on the point cloud registration after applying the predicted rotation. All other experiment details remain the same. From Table 5 we see that SVD-Train performs better than all the other baselines.

### 4.4.2 Inverse kinematics

Our second unsupervised experiment is the human pose inverse kinematics experiment [57]. A network is given 3D joint positions and is asked to predict the rotations from a canonical "T-pose" to the input pose. Predicted rotations are transformed back to joint positions via forward kinematics, and the training loss is on the reconstructed joint positions. We use the experiment code provided with [57]. Table 6 shows that SVD-Train, QCQP, and 6D all have similar performance.

### 4.4.3 Unsupervised depth estimation

The final experiment considers self-supervised learning of depth and ego-motion from videos [55]. Given a target image and source images, the model predicts a depth map for the target, and camera poses from the target to sources. Source images are warped to the target view using the predicted poses, and a reconstruction loss on the warped image supervises training. In [55] the rotational component is parameterized by Euler angles. Following [55], we report the single-view depth estimation results

Table 6: **Human pose inverse kinematics**. Following [57], we show errors in predicted joint locations in cm. Left: test errors after training 1.9M steps. Middle: errors while training progresses. Right: percentile errors after training completes.

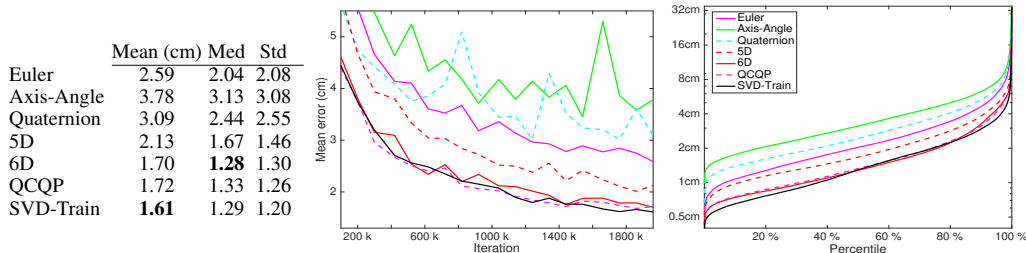

|  | Mean (cm) | Med | Std |
|---|---|---|---|
| Euler | 2.59 | 2.04 | 2.08 |
| Axis-Angle | 3.78 | 3.13 | 3.08 |
| Quaternion | 3.09 | 2.44 | 2.55 |
| 5D | 2.13 | 1.67 | 1.46 |
| 6D | 1.70 | **1.28** | 1.30 |
| QCQP | 1.72 | 1.33 | 1.26 |
| SVD-Train | **1.61** | 1.29 | 1.20 |

Table 7: **Single view depth estimation on KITTI.** We report the same metrics as in [55].

|  | Error metric ↓ | | | | Accuracy metric ↑ | | |
|---|---|---|---|---|---|---|---|
|  | Abs Rel | Sq Rel | RMSE | RMSE log | $\delta < 1.25$ | $\delta < 1.25^2$ | $\delta < 1.25^3$ |
| Euler | 0.216 | 3.163 | 7.169 | 0.291 | 0.720 | 0.893 | 0.952 |
| Axis-Angle | **0.208** | 2.752 | 7.099 | 0.287 | **0.723** | 0.894 | **0.954** |
| Quaternion | 0.218 | 3.055 | 7.251 | 0.294 | 0.707 | 0.888 | 0.950 |
| 5D | 0.234 | 4.366 | 7.471 | 0.303 | 0.717 | 0.890 | 0.950 |
| 6D | 0.217 | 3.103 | 7.320 | 0.297 | 0.716 | 0.891 | 0.951 |
| SVD-Train | 0.209 | **2.517** | **7.045** | **0.286** | 0.715 | **0.895** | 0.953 |

on KITTI [8] after 200K steps (Table 7). The error metrics are in meters while accuracy metrics are percentages up to a distance threshold in meters (see [6] for a description).

Observe that the difference between the best and second best method in each metric is small. This is not surprising since the camera pose is a small (albeit important) part of a complex deep architecture. Nonetheless, SVD-Train performs best for 4 out of the 7 metrics, and second best in another two. For driving data the motion is likely to be mostly planar for which axis-angle is well suited. Finally, it is worth noting that carefully selecting the rotation representation is important even in more complex models – the default selection of Euler angles in [55] is outperformed in every metric.

## 5 Conclusion

The results of the previous sections are broad and conclusive: a continuous 9D unconstrained representation followed by an SVD projection onto $SO(3)$ is consistently an effective, and often the state-of-the-art, representation for 3D rotations in neural networks. It is usable in a variety of application settings including without supervision, and it is easily implemented in modern machine learning frameworks. The strong empirical evidence is supported by a comprehensive theoretical analysis.

## 6 Broader impact

This work considers the a fundamental question of how to best represent 3D rotation matrices in neural networks. This is a core component of many 3D vision and robotics deep learning pipelines, so any broader impact will be determined by applications or research that integrate our proposal into their systems.

## Footnotes

[1]If $f$ is "discontinuous on a set $S$" of measure 0, it is equivalently "continuous on $\mathbb{R}^n \setminus S$."

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
