[Supplementary Material]

Supplemental Material for
# An Analysis of SVD for Deep Rotation Estimation

Jake Levinson, Carlos Esteves, Kefan Chen, Noah Snavely, Angjoo Kanazawa, Afshin Rostamizadeh, and Ameesh Makadia

## A    Complete proof of Proposition 1

In the main paper, we gave the derivative of the orthogonalization operators $\texttt{SVDO}(M)$ and $\texttt{GS}(M)$ and the resulting error under Gaussian noise, near the identity matrix $M = I$. We now give the complete proof and discussion of Proposition 1 and the additional facts about smoothness of $\texttt{SVDO}(M), \texttt{SVDO}^+(M)$.

Note that since $\texttt{SVDO}(RM) = R \cdot \texttt{SVDO}(M)$ and $\texttt{GS}(RM) = R \cdot \texttt{GS}(M)$ for any orthogonal matrix $R$, and likewise for $\texttt{SVDO}^+, \texttt{GS}^+$ if $R$ is *special* orthogonal. Therefore the error analyses are the same for all matrices $M$:

$$\|\texttt{GS}(R + \sigma N) - R\|_F^2 = \|R(\texttt{GS}(I + \sigma R^{-1}N) - I)\|_F^2 = \|\texttt{GS}(I + \sigma N) - I\|_F^2 \qquad \text{(A.1)}$$

since orthogonal matrices preserve Frobenius norm and $R^{-1}N$ has the same distribution as $N$ since $N$ was assumed isotropic. (The same applies for the other three functions.)

*Proof of Proposition 1.* (1) Let $M$ have SVD $M = U\Sigma V^T$ for some orthogonal matrices $U, V$ and diagonal matrix $\Sigma \geq 0$. To first order in $\sigma$, we can expand each of $U, \Sigma, V^T$ as

$$U = U_0(I + \sigma U_1), \qquad \text{(A.2)}$$
$$\Sigma = \Sigma_0 + \sigma \Sigma_1, \qquad \text{(A.3)}$$
$$V = V_0(I + \sigma V_1), \qquad \text{(A.4)}$$

with $U_0, V_0$ orthogonal, $U_1, V_1$ antisymmetric and $\Sigma_0, \Sigma_1 \geq 0$ diagonal. This is using the fact that the antisymmetric matrices give the tangent space to the orthogonal matrices. Similarly, the tangent space to the diagonal matrices is given again by the diagonal matrices. This gives an overall expression for $M$ as

$$M = I + \sigma N = U_0(I + \sigma U_1)(\Sigma_0 + \sigma \Sigma_1)(I + \sigma V_1)^T V_0^T. \qquad \text{(A.5)}$$

Setting $\sigma = 0$ we see $I = U_0 \Sigma_0 V_0^T$, which implies $\Sigma_0 = I$ and $U_0 = V_0$. Next, collecting the first-order $\sigma$ terms gives

$$N = U_0(U_1 + \Sigma_1 + V_1^T)U_0^T. \qquad \text{(A.6)}$$

If a matrix $X$ is (anti-)symmetric and $Q$ is orthogonal, then $QXQ^T$ is again (anti-)symmetric. So, the symmetric and antisymmetric parts of the equation are

$$S = U_0 \Sigma_1 U_0^T, \quad A = U_0(U_1 + V_1^T)U_0^T. \qquad \text{(A.7)}$$

Note that the first equation is an SVD of the symmetric part of $N$, while the second equation shows that $U_1$ and $V_1$ satisfy $U_1 + V_1^T = U_0^T A U_0$. Finally, dropping the $\Sigma_0 + \sigma \Sigma_1$ factor from Eq. (A.5) and expanding out shows that $\texttt{SVDO}(I + \sigma N) = I + \sigma A + O(\sigma^2)$.

(2) Let $M = QR$, where $Q$ is orthogonal and $R$ is upper-triangular with positive diagonal. As above, by expanding to first order in $\sigma$ we have

$$I + \sigma N = Q_0(I + \sigma Q_1)(I + \sigma R_1)R_0, \qquad \text{(A.8)}$$

with $Q_0$ orthogonal, $Q_1$ antisymmetric, and $R_1, R_0$ upper triangular. Setting $\sigma = 0$, we see $I = Q_0 R_0$ and so $Q_0 = R_0 = I$. For the $\sigma$ terms, we split $N$ into its upper, lower and diagonal parts to get

$$U + D + L = Q_1 + R_1, \qquad \text{(A.9)}$$

which by comparing parts gives $Q_1 = L - L^T$ and $R_1 = U + D + L^T$. Then $\texttt{GS}(M) = I + \sigma(L - L^T)$ by simple algebra.

We now prove Corollary 1.

**Corollary 1 (restated).** If $N$ is 3x3 with i.i.d. Gaussian entries $n_{ij} \sim \mathcal{N}(0,1)$, then with error of order $O(\sigma^3)$,

$$\mathbb{E}[\|\mathit{SVDO}(M) - I\|_F^2] = 3\sigma^2, \qquad \mathbb{E}[\|\mathit{GS}(M) - I\|_F^2] = 6\sigma^2 \qquad (A.10)$$

$$\mathbb{E}[\|\mathit{SVDO}(M) - M\|_F^2] = 6\sigma^2, \qquad \mathbb{E}[\|\mathit{GS}(M) - M\|_F^2] = 9\sigma^2 \qquad (A.11)$$

*Proof.* Simplifying the error expressions using the first-order calculations in the Proposition gives

$$\|\mathtt{SVDO}(M) - I\|_F^2 = \|\sigma A\|_F^2, \qquad (A.12)$$

$$\|\mathtt{GS}(M) - I\|_F^2 = \|\sigma(L - L^T)\|_F^2, \qquad (A.13)$$

$$\|\mathtt{SVDO}(M) - M\|_F^2 = \|-\sigma S\|_F^2, \qquad (A.14)$$

$$\|\mathtt{GS}(M) - M\|_F^2 = \|-\sigma(U + D + L^T)\|_F^2, \qquad (A.15)$$

with notation for $S, A, U, D, L$ as in the proposition. Thus each expression is $\sigma^2$ times the Frobenius norm of the corresponding matrix. Each entry of $A, L - L^T, S$ and $U + D + L^T$ is a linear combination of the entries of $N$, hence is Gaussian since $N$ has i.i.d. Gaussian entries $n_{ij} \sim \mathcal{N}(0,1)$. The expectations are the sums of the entrywise expectations of these matrices. For example, $A = \frac{1}{2}(N - N^T)$ has six nonzero entries of the form $\frac{1}{2}(n_{ij} - n_{ji})$, each having variance $\frac{1}{2}$, so $\mathbb{E}[\|A\|_F^2] = 3$. For $L - L^T$, the above diagonal entries are $-n_{ji}$ and the below-diagonal entries are $n_{ij}$, and the diagonal is 0, so the total variance is 6. The other two calculations are similar (the entries do not all have the same variances).

**Remark.** The tangent space to the identity matrix along the orthogonal matrices is the space of antisymmetric matrices. Both of the calculations above can be thought of as giving orthogonal approximations of the form

$$I + \sigma N \approx I + \sigma A', \qquad (A.16)$$

where $A'$ is a choice of antisymmetric matrix that depends on the approximation method. The fact that $\mathtt{SVDO}(M)$ produces the approximation $A' = A = \frac{1}{2}(N - N^T)$ means it corresponds to the natural projection of $N$ onto the orthogonal tangent space. By contrast, $\mathtt{GS}(M)$ produces $A' = L - L^T$, essentially a "greedy" choice with respect to the starting matrix (minimizing the change to the leftmost columns). For certain matrices $\mathtt{GS}$ can have smaller error: for example if $N$ happens to be upper-triangular, $\mathtt{GS}(M) = I$ and the error is zero. For isotropic noise, however, the SVD approximation is the most efficient in expectation.

Finally, we discuss why the error analysis is identical for $\mathtt{SVDO}^+, \mathtt{GS}^+$.

**Proposition 2** *The statements in Proposition 1 and Corollary 1 apply also for $\mathit{SVDO}^+, \mathit{GS}^+$.*

*Proof.* In Proposition 1, the matrix $N$ is fixed, so for sufficiently small $\sigma$, $\det(M) > 0$ and so $\mathtt{SVDO}(M) = \mathtt{SVDO}^+(M)$ and $\mathtt{GS}^+(M) = \mathtt{GS}(M)$.

For Corollary 1, $N$ is not fixed, so there is in fact a finite (positive) probability that $\det(M) < 0$, dependent on $\sigma$. However, the difference decays rapidly enough as $\sigma \to 0$ that the error analysis is unaffected. For any function $f(M)$, we may write $\mathbb{E}[f(M)] = (A) + (B)$, where

$$(A) = \mathrm{Prob}(\det(M) > 0) \cdot \mathbb{E}[f(M) \mid \det(M) > 0]$$
$$(B) = \mathrm{Prob}(\det(M) < 0) \cdot \mathbb{E}[f(M) \mid \det(M) < 0].$$

If $f(M)$ is the difference in error analyses between $\mathtt{SVDO}$ and $\mathtt{SVDO}^+$,

$$f(M) = \|\mathtt{SVDO}^+(M) - I\|_F^2 - \|\mathtt{SVDO}(M) - I\|_F^2,$$

then the term (A) vanishes and the term (B) is bounded by a constant times $\mathrm{Prob}(\det(M) < 0)$ since $SO(n)$ is compact. The same applies for each other comparison. Thus it suffices to show that this probability decays sufficiently rapidly. In fact, by the standard statement below, it decays faster than any polynomial, since $M$ is a Gaussian random matrix with mean $I$ and $\det(I) = 1 > 0$.

**Proposition 3** *Let $x \in \mathbb{R}^n$ be a Gaussian random vector, $x \sim N(\mu, \sigma \cdot \Sigma)$. Let $U$ be any open set containing $\mu$. Then as $\sigma \to 0$,*

$$\mathrm{Prob}(x \notin U) = O(\exp(-\tfrac{nC}{\sigma^2}))$$

*for some constant $C > 0$. In particular the decay is faster than any polynomial.*

*Proof.* The statement is invariant under affine change of coordinates, so we may assume $\mu = 0$ and $\Sigma$ is the identity matrix. Replacing $U$ by a sufficiently small unit cube around 0, the calculation factors as a product of one-dimensional Gaussians, reducing to the case $n = 1$. Rescaling by a constant $C$, we are left with calculating $\text{Prob}(|x| > 1)$ where $x \sim N(0, \sigma)$. By standard concentration inequalities, this quantity is $O(\exp(-\frac{1}{\sigma^2}))$.

## A.1 Accuracy of error estimates as $\sigma$ increases

From Corollary 1 (Sec 3.3) we see special-orthogonalization with Gram-Schmidt ($\texttt{GS}^+$) produces twice the error in expectation as SVD ($\texttt{SVDO}^+$) for $SO(3)$ reconstruction when inputs are perturbed by Gaussian noise. We compare these derived errors with numerical simulations. See Figure A1.

Figure A1: **Simulations.** We plot our derived approximations against numerical simulations of the expected error in reconstruction under additive noise. For each $\sigma$ we compute the numerical expectation with 100K trials. These plots can provide a sanity check of our derivations.

# B   Proof of smoothness and discussion

**Proposition 4** *The symmetric orthogonalization* $\texttt{SVDO}(M)$ *is a smooth function of* $M$ *if* $\det(M) \neq 0$.

*Proof.* We use the Implicit Function Theorem and the least-squares characterization of $\texttt{SVDO}(M)$ as

$$\texttt{SVDO}(M) = \underset{Q \in O(n)}{\arg\min} \|M - Q\|_F^2. \tag{B.1}$$

We calculate the derivative with respect to $Q \in O(n)$: for $A$ an antisymmetric matrix,

$$\lim_{\epsilon \to 0} \tfrac{1}{\epsilon}(\|M - Q(I + \epsilon A)\|_F^2 - \|M - Q\|_F^2) = -2\,\text{Tr}(M^T Q A). \tag{B.2}$$

If this vanishes for every $A$, then $M^T Q$ is symmetric, that is, $(M, Q)$ is a root of the function $g(M, Q) = M^T Q - Q^T M$. Let $M_0$ be a fixed matrix. As discussed above, the optimal solution to this equation is given by an SVD, $M_0 = U_0 S_0 V_0^T$, yielding $Q_0 = U_0 V_0^T$. To show that $Q$ is a smooth function of $M$, it suffices by the Implicit Function Theorem to show that the Jacobian matrix $\frac{\partial g}{\partial Q}$ is full-rank at $(M_0, Q_0)$. To see this, we differentiate it again:

$$\frac{\partial g}{\partial Q}(A) = \lim_{\epsilon \to 0} \frac{1}{\epsilon}(g(M_0, Q_0(I + \epsilon A)) - g(M_0, Q_0)) = M_0^T Q_0 A - A^T Q_0^T M_0, \tag{B.3}$$

where $A$ is antisymmetric. Some algebra shows that this is, equivalently,

$$\frac{\partial g}{\partial Q}(A) = V_0(S_0 V_0^T A V_0 + V_0^T A V_0 S_0)V_0^T. \tag{B.4}$$

To see that this is an invertible transformation of $A$, note that conjugating by $V_0$ is invertible since $V_0$ is orthogonal. So it is equivalent to show that the function

$$A \mapsto S_0 A + A S_0 \tag{B.5}$$

is invertible. This function just rescales the entry $a_{ij}$ to $(s_i + s_j)a_{ij}$. Since the singular values are positive this is invertible as desired.

**Proposition 5** *The special symmetric orthogonalization is a smooth function of $M$ if either of the following is true:*

- $\det(M) > 0$,

- $\det(M) < 0$ *and the smallest singular value of $M$ has multiplicity one.*

*Proof sketch.* The analysis is identical to the main proof, except that if $\det(M) < 0$, $S_0$ is effectively altered so that the last entry is changed from $s_n$ to $-s_n$. Thus the function $A \mapsto S_0 A + A S_0$ now sends the entry $a_{ij}$ to $(\pm s_i \pm s_j) a_{ij}$, with negative signs at $i = n$ and/or $j = n$. If $s_n$ occurred with multiplicity one, the result is still invertible since $s_i - s_n \neq 0$ for $i \neq n$ and for $i = j = n$ the coefficient is $-2s_n$. Otherwise, however, $s_{n-1} = s_n$ and the operation sends $a_{n-1,n}$ to $(s_{n-1} - s_n) a_{n-1,n} = 0$; likewise for $a_{n,n-1}$. In this case there are many optimal *special* orthogonalizations of $M_0$, and the operation is not even continuous in a neighborhood of $M_0$.

## C Gradients

Here we provide the details for the derivation sketched out in Sec 3.2, which analyzes the behaviour of the gradients of a network with an $\mathtt{SVDO}^+$ layer. Specifically, we consider $\frac{\partial L}{\partial M}$ for some loss function $L(M, R) = \|\mathtt{SVDO}^+(M) - R\|_F^2$.

We will first analyze $\frac{\partial L}{\partial M}$ for $\mathtt{SVDO}(M)$. With $\circ$ denoting the Hadamard product, from [18, 48] we have

$$\frac{\partial L}{\partial M} = U[(F^T \circ (U^T \frac{\partial L}{\partial U} - \frac{\partial L}{\partial U}^T U))\Sigma + \Sigma(F^T \circ (V^T \frac{\partial L}{\partial V} - \frac{\partial L}{\partial V}^T V))]V^T, \quad \text{(C.1)}$$

$$F_{i,j} = \begin{cases} \frac{1}{s_i^2 - s_j^2}, & \text{if } i \neq j \\ 0, & \text{if } i = j \end{cases}, \quad s_i = \Sigma_{ii}. \quad \text{(C.2)}$$

Let $X = U^T \frac{\partial L}{\partial U} - \frac{\partial L}{\partial U}^T U$, and $Y = V^T \frac{\partial L}{\partial V} - \frac{\partial L}{\partial V}^T V$. Since $\|\mathtt{SVDO}(M) - R\|_F^2 = 2\,\text{Tr}(\mathbb{I}_n) - 2\,\text{Tr}(UV^T R^T)$, then $\frac{\partial L}{\partial U} = -2RV$, and $\frac{\partial L}{\partial V} = -2R^T U$. This leads directly to $X = Y^T = -Y$ ($X, Y$ are antisymmetric). We can simplify Eq. C.1 as

$$\frac{\partial L}{\partial M} = U((F^T \circ X)\Sigma - \Sigma(F^T \circ X))V^T. \quad \text{(C.3)}$$

Inspecting the individual elements of $(F^T \circ X)\Sigma$ and $\Sigma(F^T \circ X))$ we have

$$\left((F^T \circ X)\Sigma\right)_{ij} = \frac{X_{ij}s_j}{s_j^2 - s_i^2}, \qquad \left(\Sigma(F^T \circ X)\right)_{ij} = \frac{X_{ij}s_i}{s_j^2 - s_i^2}. \quad \text{(C.4)}$$

Letting $Z = (F^T \circ X)\Sigma - \Sigma(F^T \circ X)$, we can simplify $\frac{\partial L}{\partial M} = UZV^T$ where the elements of $Z$ are

$$Z_{ij} = \begin{cases} \frac{X_{ij}}{s_i + s_j}, & \text{if } i \neq j \\ 0, & \text{if } i = j. \end{cases} \quad \text{(C.5)}$$

For $\mathtt{SVDO}(M)$ Eq. C.5 tells us $\frac{\partial L}{\partial M}$ is undefined whenever two singular values are both zero and large when their sum is very near zero.

For $\mathtt{SVDO}^+(M)$, if $\det(M) > 0$ then the analysis is the same as above. If $\det(M) < 0$, the extra factor $D = \text{diag}(1, 1, \ldots, -1)$ effectively changes the smallest singular value $s_n$ to $-s_n$. The derivation is otherwise unchanged. In particular the denominator in equation (C.5) is now $s_j - s_n$ or $s_n - s_i$ if either $i$ or $j$ is $n$.

### C.1 Gradients observed during training

In Figure A2 (left) we see the gradient norms observed while training for point cloud alignment (Section 4.1). SVD-Train has the same profile as for 6D ($\mathtt{GS}^+$). SVD-Train converges quickly (relative to all other methods) in all of our experiments, indicating no instabilities due to large gradients.

On the right of Figure A2 we profile the gradients for the scenario where we begin training with the SVD-Inference loss and switch to SVD-Train after 100K steps (after roughly 4% of training iterations). SVD-Inf trains the network to produce outputs that are close to $SO(3)$, which eliminates some conditions of instability in Eq. C.5. This is confirmed by seeing much smaller gradient norms after switching to SVD-Train at 100K steps. Note, this approach was never used (or needed) in our experiments.

Figure A2: **Gradients.** Left are the gradient norms $\|\frac{\partial L}{\partial M}\|_F^2$ for the point cloud alignment experiment. SVD-Train and 6D have similar profiles. On the right the network is trained with SVD-Inf for the first 100K steps, then SVD-Train. During the first 100K steps the network learns to output matrices close to $SO(3)$ and this eliminates sources of high gradient norms in Eq. C.5.

# D   Experiments

## D.1   Additional baseline details

- **Spherical Regression** ($S^2$-**Reg**) regresses to $n$-spheres. Following [25], we use regression to $S^1$ for Pascal3D+ and $S^3$ regression (quaternions) for their ModelNet experiments (section 4.2). The method combines abs. value regression with sign classification. Our implementation of the final regression layers follows the provided code. We select the hyperparameter that balances the classification and regression losses by a simple line search in the neighborhood of the default provided in [25].

  $S^2$-Reg uses both classification and regression losses, not surprisingly we were unable to train successfully on any of the unsupervised rotation experiments. The closest we came was on unsupervised point cloud alignment (Sec 4.4). With careful hyperparameter tuning the model completed training with mean test errors near $90°$.

- **3D-RCNN** [23] combines likelihood estimation and regression (via expectation) for predicting Euler angles. This representation also requires both classification and regression losses for training, and we were unable to make the model train successfully on the unsupervised rotation experiments.

- **Geodesic-Bin-and-Delta** ($\mathbf{M}_G$ [28]) combines classification over quantized pose space (axis-angle representation) with regression of offsets. For our experiments with where observed rotations are uniformly distributed over $SO(3)$ (Sec. 4.1, 4.2), $K$-means clustering is ineffective. Instead we quantize $SO(3)$ by uniformly sampling a large number (1000) of rotations (larger values did not improve results). We found this version of Geodesic-Bin-and-Delta outperformed the One-delta-network-per-pose-bin variation in these experiments. For Pascal3D+ we follow the reference and use $K$-means with $K = 200$. This method also requires both classification and regression losses and we were unable to train successfully in the unsupervised setting.

- **Quaternion**, **Euler angles (XYZ)**, and **axis-angle** are the classic parameterizations. In each case they are converted to matrix form before the loss is applied. In our experiments we did not filter any outputs from the network representing angles (e.g. clipping values or applying activations such as $\tanh$). We found this gave the best results overall.

## D.2 Learning rate decay

An observation from the point cloud registration results is that the curves for mean test errors as training progresses do not decay smoothly as one might expect for any method (Table 1, middle, in the main paper). This is in part due to the training code from [57] does not utilize a learning rate decay for this experiment. It is reasonable to observe the variance in evaluation when a decay is introduced as would be common in practice. Table A1 (left) shows the curves when the learning rate is exponentially decayed (decay rate of 0.95, decay steps of 35K). The evaluation over time is smoother but the results are consistent with those presented in the main paper. Table A1 (right) shows SVD-Train performance with three different initial learning rates with decay. The higher learning rate of $1e-4$ improves performance (Mean/Med errors of $1.32°/0.60°$), indicating the benchmark performance could be improved with hyperparmeter tuning (likely for all methods).

Table A1: **Left: Point cloud alignment with learning rate decay**. Evaluation is smoother over time but the comparative analysis does not change. **Right: Different learning rates for SVD-Train.**

## D.3 Geodesic loss

In [57] it was shown that geodesic loss for training did not significantly alter the results, and we confirm this observation in Table A2 (left).

## D.4 2D Pose

In Table A2 (right), we replicate the point cloud alignment experiment while restricting the rotations to a 2D subspace of $SO(3)$ that can be identified with the 2-sphere.

Table A2: **Left:** Training with geodesic loss for point cloud alignment. Relative performances are consistent with squared Frobenius loss (Table 1 in main paper). **Right:** point cloud alignment when training and test rotations are restricted to a 2D subspace of $SO(3)$.

| Geodesic loss | | | |
|---|---|---|---|
| | Mean (°) | Med | Std |
| 5D | 3.88 | 2.08 | 9.19 |
| 6D | 2.29 | 1.30 | 7.52 |
| SVD-Train | 2.05 | 1.28 | 7.15 |

| 2D Pose | | | |
|---|---|---|---|
| | Mean (°) | Med | Std |
| 6D | 0.89 | 0.44 | 4.60 |
| SVD-Train | 0.64 | 0.31 | 4.88 |

## D.5 Pascal3D+ full results.

Here in Table A3 we show the results for all 12 categories in the Pascal3D+.

Table A3: **Pascal 3D+.** Results for all 12 categories.