[Reviews · NeurIPS 2020]

Review 1

Summary and Contributions: This paper presents an analysis of various parameterizations of 3D rotations, and (rather surprisingly) deduce that using SVD to orthogonalize a matrix is the best estimator for tasks involving neural networks.

Strengths: [S1] This paper (re-)revisits a central question in 3D computer vision: "how should one represent rotations for use with neural networks"? Several approaches over the last 3-4 years have argued for a completely different representation, each time claiming their method performs the best. It is a welcome effort that the authors of the current manuscript have undertaken, to evaluate all these representation choices on equal footing. [S2] SVD as a tool for rotation estimation has been overlooked (quoted from the paper) over the years. I enjoyed reading through the technical proofs on why the paper posits "SVD-plus" as an elegant solution. In particular, I liked the discussion on perturbing small matrices for stable derivatives. [S3] The experimental analysis includes a wide range of tasks (3D pose estimation, pointcloud alignment, inverse kinematics learning, depth estimation, and more). In all of their experiments, and nearly under all metrics, SVD-plus performs better compared to evaluated alternate representations.

Weaknesses: [W1] I am quite surprised by the result (understandably, as many others). In the 3D computer vision and robotics communities, there has always been a preference for Lie groups to characterize 3D transformations [R1, R2]. While the SVD parameterization is appealing because of it being a global representation, the paper does not seem to investigate why SVD-plus is better (albeit for a comparison with [47] in Corrolary 3). [W2] For Euler angles in particular, it is prudent to know of the parameterization being applied. While the Cayley parameterization is suited for problems where the initial guess is within the convergence basin of the global optimum, it fails to converge when the initial and true rotations are far apart. A more careful parameterization of the SO(3) Jacobians, as outlined in [R1], [R2] should be considered for a fair comparison. [W3] Line 110 assumes that the noise distribution over M is Gaussian. The reason behind that is not entirely clear. It has been discussed, for example in [7], and [R3] that Bingham and Langevin distributions are better suited to model errors over SO(3). While a Langevin distribution can "loosely" be considered to be an analogue of a Gaussian in Euclidean space, it would help to think deeper about this assumption (as the comparison with [47] rests on this). [W4] One aspect that's unclear from the empirical analysis is whether this idea would hold for special cases of rotations (eg. about a fixed axis, about a fixed plane of revolution, etc.). While the theoretical analysis suggests that SVD-plus works for such cases, it'll be interesting to asses if the trends in performance changes over time.

Correctness: To the best of my knowledge and abilities, the method, evaluation protocols are correct. Wherever I've found the empirical methodology lacking, I've outlined it as a corresponding "weakness".

Clarity: The paper is extremely well-written. No complaints whatsoever!

Relation to Prior Work: I think the paper does a fair characterization of related work. There's some literature in the control theory community that attempted to carry out similar analyses, and they are worth citing. See [R4], for starters.

Reproducibility: No

Additional Feedback: I wonder if this approach can be extended to other kinds of 3D transformations, such as rototranslations (i.e., the SE(3) group), or scale-ambiguous rototranslations (i.e., the Sim(3) group). While SVD might not directly apply, there's lots more work to be done on SE(3) and Sim(3) in terms of analyzing the impact of choice of representation. As an aside, I wonder if the current techniques "generalize" to SE(2) (i.e., 2D rotations). There could be interesting insights to draw from SE(2). [R1] A micro Lie theory for state estimation in robotics. [R2] State estimation for robotics. Tim Barfoot. [R3] SE-Sync: A Certifiably Correct Algorithm for Synchronization over the Special Euclidean Group. [R4] On Attitude Representations for Optimization-BasedBayesian Smoothing. UPDATE: Upon reading the author response and the other reviews, I find most of my concerns adequately addressed. I tend to respectfully disagree with some of the other reviewers that the paper lacks novelty. The result of this paper, in my opinion, is surprising, strong, and thus novel. My updated score reflects this.


Review 2

Summary and Contributions: *** Post rebuttal update *** I have read the authors' response and considered the modifications they'll introduce based on the feedback. Despite having read all the reviews and the rebuttal, I continue to have serious concerns about this paper and upon further investigation after reading the rebuttal I would like to clarify why I am reducing my score: * Proposition 1 + Corollary 1: These are very poorly phrased and the proofs are quite haphazardly written, e.g. matrix dimensions are missing and it's not clear what 'first order' even means. Proposition 1 is also quite uninteresting and it's not clear what its significance is; the 'proof' is trivial. * Supplement: the two proofs of the two propositions on smoothness (which are only mentioned in line 171 in the main paper without delineating them as standalone propositions): the smoothness results (i.e. SVD minimizes the Frob norm) is a straightforward proofs present in any advanced linear algebra textbook. * Line 135-136: the statement "arbitrary (non-random)" sounds confusing and also the whole sentence is not actually grammatically correct. * As mentioned in my review L174–176 are trivial and although the authors say they will update the paper to reiterate the least-squares optimality of SVD is well-known, this will not actually alleviate my concern: simply re-stating well known results from numerical linear algebra will not make the paper better suited for this conference. * Per authors' own words: "Note, we do not claim to be the first to introduce this tool to deep learning, rather our focus is on providing a comprehensive study of the technique specifically for estimating rotations. Our contributions include a theoretically motivated analysis" (lines 44-45). Based on what I wrote above, I have serious concerns about their 'theoretically motivated' analysis and do not see either the algorithmic or theoretical contribution of this paper. ********************************************** The paper begins with a statement that symmetric orthogonalization via SVD has been underused in deep learning for pose estimation and related computed vision problems. The authors define several variations of SVD, define how it can be used for matrix projection in pose estimation and provides an experimental study to illustrate its usefulness for estimating rotations in pose estimation and related compute vision applications. The main contribution of the paper is empirical evaluation of several methods with a focus on the SVD approach (which is well studied in the literature but not well explored in the specific setting of pose estimation).

Strengths: The paper provides a fairly thorough empirical study with a rich set of experiments.

Weaknesses: While an interesting read, the paper is not of high relevance to the NeurIPS community due to the lack of groundbreaking result or an unexpected and novel empirical observation of a phenomenon. It is overall not surprising that in fact SVD projection works well in the low dimensional setting described in the paper. While its underutilization in deep learning application for pose estimation might be concerning and even surprising, I think this is more of an artifact of the incentives that exist in computer vision research in general, rather than due to some surprising 'properties' of 3D pose estimation problems. -- Lines 95-98 are repeats of lines 89-92; please fix this -- Lines 174-176: well known result albeit underutilized in deep learning applications.

Correctness: I did not find inaccurate claims, although I have some concerns highlighted in the "Weaknesses" section above.

Clarity: Overall the paper meets the bar in terms of clarity.

Relation to Prior Work: The paper discusses related prior work on SVD and its use for a variety of application in linear algebra to neural nets.

Reproducibility: Yes

Additional Feedback: The authors provide code for the SVD projection from 9D to 3D but do not provide the code their experiments are based on. In any event, this is a minor issue.


Review 3

Summary and Contributions: In this work, the authors present study of different 3D rigid body rotation representations in the context of deep neural networks. They argue SVD orthogonalization (i.e., the projection of an arbitrary 3x3 matrix onto the special orthogonal group, SO(3)) has a set of particularly attractive properties in this domain. Their main contributions are the theoretical analysis of SVD orthogonalization in the context of deep rotation learning, and quantitative comparison of this representation with other classical approaches (axis-angle, quaternions and Euler angles) as well as some recently-suggested representations (e.g., a 6-parameter representation that uses Gram-Schmidt for orthogonalization).

Strengths: This is a well-written paper with both sounds theoretical claims and thorough empirical validation. The general question of rotation representation is a very important design decision in many learned and classical state estimation pipelines. This has broad applicability to many NeurIPS-related subject areas within robotics and computer vision. The authors outline a number of attractive properties of their SVD-based representation with clear, cogent mathematical derivations and easy-to-read prose. Their experiments build on prior work and facilitate direct comparisons with competing approaches.

Weaknesses: First, it is not immediately clear if the continuity described in section 3.4 is the same type of 'global right-inverse' continuity described in Zhou et al. ([47] in the paper). This should be clarified. Second, the authors state that probabilistic representations 'do not reach state of the art in tasks where a single precise rotation must be predicted'. However, recent work [1] that presents a rotation representation based on a symmetric 4x4 matrix, with 10 parameters (which can be mapped to a SO(3) via an eigendecomposition and which also defines a Bingham density) may be at odds with this statement. Although this work was published contemporaneously with the submission of this work, given its similarity and relevance it would be interesting to see it added to the quantitative analysis of this work. Finally, the experiments in Tables 1-4 show learning curves for single optimization 'runs'. It is not clear how sensitive these trends are to learning rates and other optimization hyper-parameters. [1] Peretroukhin et al. A Smooth Representation of Belief over SO(3) for Deep Rotation Learning with Uncertainty, Robotics: Science and Systems (RSS) 2020

Correctness: In general, yes. One specific qualm: In section 4.1, line 207, the authors state that the maximum error for all methods was approximately 180 degrees. If this section reproduces the 'ShapeNet' experiments of [47] which use airplane point clouds (this fact itself should be clarified within the paper), then it is not clear to me that such large errors are simply a function of data symmetry. How does the symmetries manifest itself in airplane point-clouds? The work cited above ([1]) argues that large errors are primarily due to representation discontinuities.

Clarity: The prose of this paper is clear and very well proof-read (I did not spot a single grammatical error). Some minor points: Line 226: It would be useful here to explain what it means to sample uniformly over SO(3). Are you sampling in axis-angle space? Lines 199-200: This chordal metric is related to the geodesic metric as you state (and also the quaternion metric -- See Hartley et al., Rotation Averaging, IJCV (2013)). Did you find there to be any reason to use the chordal loss over others in the context of supervised learning? This may be worth clarifying at this point.

Relation to Prior Work: Well-situated within the current literature on deep learning that involves rotations, with the exception of a recent RSS paper (see Weaknesses) which presents a novel representation that the authors are encouraged to add to their analysis.

Reproducibility: Yes

Additional Feedback: ---- UPDATE ----- I thank the authors for their detailed rebuttal. Their updated experiments incorporating a recently-published (and similar-in-spirit) rotation representation will make this work even more compelling. I maintain my rating - I think this is an excellent piece of work.


Review 4

Summary and Contributions: The main contributions of this paper are twofold: 1) A mathematical analysis of rotation estimation using SVD decomposition (SVD orthogonalization and SVD special orthogonalization). In particular, the authors show that SVD orthogonalization improves expected error over the Gram-Schmidt method by a factor of 2. 2) An extensive set of experiments that show the state-of-the-art level performance of this method for estimating rotation matrices in the context of neural networks.

Strengths: Main strengths: * Presentation. The paper is extremely well-written. The claims are clearly stated, and sections 3 and 4 provide very compelling evidence for their validity, both mathematical and experimental. * Experiments. The performance evaluation of the method presented in the paper is extremely thorough. The proposed orthogonalization technique is tested in about half-a-dozen scenarios and datasets, and compared to several competing algorithms, achieving state-of-the-art performance in all experiments.

Weaknesses: Main weaknesses: * Novelty. The main concern I have is about novelty, i.e., whether the contributions of the paper on their own warrant publication in this venue. * Details. Another minor weakness is the lack of details in some sections of the paper. In particular, section 4 is missing details about loss functions, benchmarks, and even some results (table 4), all of which are relegated to the supplemental materials.

Correctness: Yes. The paper's main claim -- that rotation matrix estimation in deep learning via SVD orthogonalization -- is very solidly backed up by mathematical analysis, as well as by an extensive quantitative evaluation. In particular, the method is tested on a very diverse set of applications and datasets, and compared against many state-of-the-art methods. The math also seems correct, although I did not check in full detail (especially the supplemental material).

Clarity: Yes. The paper is very well written. The story of the paper is very simple and straightforward, and the paper reads very naturally. The only concern is about some lack of detail. As mentioned above, I would llke to see more details of the experimental setup make it into the paper, rather than be relegated to the supplemental material.

Relation to Prior Work: Although I am not an expert in this area, I found the discussion on related work satisfactory. In particular, the authors make it a point to distinguish their results from the state-of-the-art, all of which are briefly discussed in 2.

Reproducibility: Yes

Additional Feedback: Regarding the novelty concern above, it would be interesting to see how using this approach to rotation estimation impacts other "downstream" computations or layers in a NN. That is, does the reduction in error actually have a significant impact on networks that will use this technique? Regarding missing details, I would suggest to the authors to remove the proof sketch in of proposition 1, and use the space to fill in more details about the experiments. The proof sketch does not provide much intuition, and certainly does not constitute an actual proof, so it is better to omit it in benefit of other details.

[Author Response · NeurIPS 2020]

**Author response for *An Analysis of SVD for Deep Rotation Estimation*, Paper ID 11801.**

We thank all the reviewers for their efforts and constructive comments, and for recognizing the contribution of our
comprehensive mathematical and experimental analysis in support of $\text{SVD}^+(M)$. We first address the concerns of
relevance and novelty mentioned by two reviewers: **[R2]** "*overall not surprising. . . an artifact of the incentives in*
*computer vision research*," and **[R4]** "*whether the contributions. . . warrant publication in this venue.*"
**Surprising:** There is ample evidence that our contribution would be received as surprising by the research community.
Deep learning research for vision/robotics applications is judged on the output (e.g. 3D reconstruction, depth estimates,
skeleton pose), not by their network's internal rotation representation. *Thus the incentive is to use the best available*
*representation*. That domain experts do not consider SVD (L40, [19,4,30,24]) indicates our results would be surprising.
This is supported by other reviewers, e.g. **[R1]** "*quite surprised by the result (understandably, as many others).*"
**Novel:** In addition to the thorough experimental analysis, the mathematical analysis is an important component of the
exposition and is also a novel contribution. The error analysis derivation (Sec 3.3, e.g. Corollary 1: $\text{SVD}^+$ error is $3\sigma$,
$\text{GS}^+$ error is $6\sigma$), the theoretical and empirical gradient analysis (Sec 3.2, Supp. 3.1), and discussion on continuity (Sec
3.4), are all novel contributions. This analysis provides the theoretical grounding supporting $\text{SVD}^+$ in neural networks.
**Relevant:** Rotation estimation in neural networks has **[R3]** "*broad applicability to many NeurIPS-related subject*
*areas*," and is **[R1]** "*a central question in 3D computer vision.*" Given the surprising and comprehensive empirical
findings, along with a novel mathematical analysis tailored for deep learning and for comparison to state-of-art methods
(SVD vs GS [47]), this work is very relevant to the NeurIPS community.

**[R3]** "*if the continuity described in section 3.4 is the same type of 'global right-inverse' continuity described in [47].*"
We use "continuity" in the conventional sense of continuous functions and differentiability. The global right-inverse
condition imposed by [47] automatically applies to our setting since our 9D representation space by definition contains
SO(3) as a subspace (and SO(3) itself is fixed by the projection functions $\text{SVD}(M)$, $\text{SVD}^+(M)$)

**[R1]** "*does not seem to investigate why SVD-plus is better (albeit for a comparison with [47] in Corrolary 3).*" Prior
work [47, 20] has carefully analyzed the limitations of classic SO(3) representations in neural networks, so we focused
our comparative analysis on $\text{GS}^+(M)$ [47] since GS is closely related to SVD and is the current state of the art. We
believe our analysis (SVD as the natural robust projection onto SO(3), stable gradients, etc) explain its success in the
experiments. We will include a discussion placing our analysis in the context of classic representations.

**[R1]** "*[L110] the noise distribution over M is Gaussian. . . Bingham and Langevin distributions are better suited to*
*model errors over SO(3).*" Here the noise model represents errors introduced by networks when predicting unconstrained
9D outputs rather than errors in SO(3). We will add the references and include a clarification discussion.

**[R3]** "*Peretroukhin et al, RSS2020. . . published contemporaneously.*" Thanks for the suggestion. Although this paper
appeared *after* the NeurIPS deadline, we will include a discussion and add it to all experiments in the final version.
Preliminary results indicate it ranks 2nd for Pt. Clouds (Table 1): mean/med err of $1.97/1.06°$ vs $1.63/0.89$ for $\text{SVD}^+(M)$.

**[R3]** L207: "*large errors. . . due to representation discontinuities.*" The ShapeNet airplanes used by [47] contain
spaceships with perfect $180°$ symmetry. We will add images to the supplemental, and rephrase the text to indicate that
in general, errors for an unseen test set can depend on representation, model generalization, and data ambiguities.

**[R1]** "*For Euler angles . . . it is prudent to know of the parameterization.*" We treat the network output as *XYZ* Euler
angles. We did not consider alternatives since we were following previously established experimental settings, e.g. [47],
but we will include alternatives (Cayley) in the final version. We thank **R1** for the references on state estimation and
control theory, and will include a discussion in our related work and analysis.

**[R1]** "*empirical analysis . . . would hold for special cases of rotations (eg. about a fixed axis. . . .*" The KITTI dataset
(Table 7) is mostly planar motion. We will add other special cases in supplemental by simulating data with 3D shapes.

**[R1]** "*if this approach can be extended to . . . SE(3) . . . Sim(3).*" $\text{SVD}^+(M)$ could be deployed in a straightforward way
for regression to product spaces involving SO(3) by simply decoupling SO(3) from the other terms (e.g. regressing $\mathbb{R}^3$
and SO(3) separately). We leave it to future work to analyze different approaches in practice.

**[R4]** how "*rotation estimation impacts other 'downstream' computations.*" Inverse Kinematics and KITTI depth (Tables
6 and 7) are examples of established applications where accurate rotation estimates impact downstream objectives.

**[R3]** "*[LR, other] hyper-parameters.*" The conclusions remained with/without LR-decay (Tab. 1 and Supp 4.2), different
losses (Supp 4.3) and encoding models (Supp 4.4.2). We will include an experiment with granular change in LR.

**Other points:** We will release the experiment code as well (**R2**). We sample random rotations according to the Haar
measure on SO(3) (**R3**). We found no change between chordal and geodesic loss (Supp Sec 4.3) (**R3**). We will
restructure the paper according to the helpful suggestion from **R4** to include more details in the main body. We will
update the analysis summary (L174–176) to reiterate the least-squares optimality of SVD is well-known. (**R2**).

[Meta-Review · NeurIPS 2020]

All reviewers concur that it is surprising that SVD as presented in this paper is not more widely used in DL applications. The primary value of the paper is really the empirical evidence that SVD is valuable. The reviewers disagree somewhat on the level of surprise in the derived results, but it remains valuable to include them in the paper, as a reference for readers who wish to make use of the empirical findings. The abstract's claim "we provide a theoretical analysis that shows SVD is the natural choice..." is rather empty. I might just as well show quaternions are "natural", by deriving some other quantities. I think it would be much better to write this as "We summarise some theoretical properties of SVD as used for representation of (and projection onto) the rotation group."